# Pentoxifylline Inhibits TNF-α/TGF-β1-Induced Epithelial-Mesenchymal Transition via Suppressing the NF-κB Pathway and *SERPINE1* Expression in CaSki Cells

**DOI:** 10.3390/ijms241310592

**Published:** 2023-06-24

**Authors:** Luis Arturo Palafox-Mariscal, Pablo Cesar Ortiz-Lazareno, Luis Felipe Jave-Suárez, Adriana Aguilar-Lemarroy, María Martha Villaseñor-García, José Roberto Cruz-Lozano, Karen Lilith González-Martínez, Aníbal Samael Méndez-Clemente, Alejandro Bravo-Cuellar, Georgina Hernández-Flores

**Affiliations:** 1Doctoral Program in Biomedical Sciences Orientation Immunology, University Center for Health Science (CUCS), University of Guadalajara (UdeG), 44340 Guadalajara, Jalisco, Mexico; luis.palafox1821@alumnos.udg.mx (L.A.P.-M.); jroberto.cruz@alumnos.udg.mx (J.R.C.-L.); 2Immunology Division, Biomedical Research Center West (CIBO), Mexican Social Security Institute, 44340 Guadalajara, Jalisco, Mexico; pablo.ortiz@imss.gob.mx (P.C.O.-L.); luis.jave@imss.gob.mx (L.F.J.-S.); adry.aguilar.lemarroy@gmail.com (A.A.-L.); mvillas@gmail.com (M.M.V.-G.); karen.gonzalez0944@alumnos.udg.mx (K.L.G.-M.); 3Department of Pharmacobiology, University Center for Exact Sciences and Engineering (CUCEI), University of Guadalajara (UdeG), 44340 Guadalajara, Jalisco, Mexico; 4Doctoral Program in Molecular Biology in Medicine, University Center for Health Science (CUCS), University of Guadalajara (UdeG), 44340 Guadalajara, Jalisco, Mexico; 5Microbiology Laboratory, Civil Hospital of Guadalajara “Fray Antonio Alcalde”, 44280 Guadalajara, Jalisco, Mexico; asamelmendez@gmail.com; 6Department of Health Sciences, Los Altos University Center (CUALtos), University of Guadalajara (UdeG), 47620 Tepatitlan de Morelos, Jalisco, Mexico

**Keywords:** pentoxifylline, epithelial-mesenchymal transition, NF-κB, *SERPINE1*, CaSki cells

## Abstract

Cervical cancer (CC) is one of the most common and deadly types of female cancer worldwide. Late diagnosis in CC increases the risk of tumor cells spreading to distant organs (metastasis). The epithelial-mesenchymal transition (EMT) is a fundamental process of cancer metastasis. Inflammation can lead to tumor progression, EMT induction, and metastasis. The inflammatory microenvironment is a potent inducer of EMT; inflammatory cytokines such as Tumor Necrosis Factor-alpha (TNF-α) and Transforming growth factor-beta (TGF-β1) activate transcriptional factors such as STAT3, Snail, Smad, and the Nuclear Factor kappa light-chain-enhancer of activated beta cells (NF-κΒ), which drive EMT. Anti-inflammatory compounds may be an option in the disruption of EMT. PenToXifylline (PTX) possesses potent anti-inflammatory effects by inhibiting NF-κB activity. In addition, PTX exerts an anti-fibrotic effect by decreasing Smad2/3/4. We hypothesize that PTX could exert anti-EMT effects. CaSki human cervical tumor cells were exposed to TNF-α 10 ng/mL and TGF-β1 alone or in combination for 5 days. Our results revealed that TNF-α and TGF-β1 induced N-cadherin and Vimentin, confirming the induction of EMT. Furthermore, the combination of cytokines synergized the expression of mesenchymal proteins, enhanced IκBα and p65 phosphorylation, and upregulated Serpin family E member 1 (*SERPINE1*) mRNA. PTX pretreatment prior to the addition of TNF-α and TGF-β1 significantly reduced N-cadherin and Vimentin levels. To our knowledge, this is the first time that this effect of PTX has been reported. Additionally, PTX reduced the phosphorylation of IκB-α and p65 and significantly decreased *SERPINE1* expression, cell proliferation, migration, and invasion. In conclusion, PTX may counteract EMT in cervical cancer cells by decreasing the NF-κB and *SERPINE1*.

## 1. Introduction

CC is the fourth most common cancer among women of reproductive age, with an estimated 341,831 deaths in 2020 [1]. Late diagnosis in CC increases the risk of tumor cells spreading to distant organs. This cellular dissemination is a process known as metastasis. Metastasis is responsible for 90% of cancer deaths. In tumor progression, the gradual loss of epithelial identity and the gain of the mesenchymal phenotype drive EMT, and in epithelial cancers, the EMT process is crucial for organ invasion [2,3]. The loss of E-cadherin and the upregulation of N-cadherin and Vimentin is the principal characteristic of EMT [4]. Inflammatory cytokines such as Tumor Necrosis Factor-alpha (TNF-α) and Transforming Growth Factor-beta (TGF-β1) activate the NF-κB and Smad pathways that drive EMT [5].

Both activated pathways result in intracellular signals of transduction that activate the genes that drive the transformation of epithelial cells into the mesenchymal type [6,7,8,9]. It has been reported that TNF-α engages in intense crosstalk with TGF-β1 to synergize EMT induction [10]. Targeting TNF-α and TGF-β1 may be a valid possibility for disrupting EMT. In this sense, molecules with anti-inflammatory properties may open new therapeutic options in the treatment of cancer.

PTX is a dimethyl xanthine proven to significantly increase apoptosis in the leukemic cells of pediatric patients with Acute lymphoblastic leukemia (ALL) during treatment with Prednisone [11] and to improve the prognosis of patients with hepatocarcinoma [12]. On its addition, PTX demonstrates an outstanding anti-inflammatory and anti-fibrotic performance [13,14,15]. The specific mechanisms behind these effects exerted by PTX may be related to the reduction of the plasma levels of proinflammatory cytokines such as IL-1, IL-6, TNF-α, and TGF-β1, the inhibition of NF-κB and Smad2/3/4 activity, and the decrease of the expression of the plasminogen activator inhibitor-1 (PAI-1) and fibronectin [16,17,18,19,20].

Therefore, we have reason to think of the practicability of PTX in disrupting EMT. This work aimed to evaluate the effect of PTX in the EMT induced by TNF-α/TGF-β1 in CaSki human cervical cancer cells and their possible mechanisms.

## 2. Results

### 2.1. Cell Viability

CaSki cells were treated with increasing concentrations of PTX (0.5, 1, 2, and 4 mM), TNF-α (5, 10, and 15 ng/mL), and TGF-β1 (5, 10, and 15 ng/mL) for 5 days. Cell viability was evaluated using 7AAD staining by flow cytometry. We can observe, in Figure 1A, the effect of PTX on CaSki cervical cancer cells; the concentrations of 0.5 and 1 mM did not affect the viability of these cells, and the concentrations of 2 and 4 mM of PTX decreased viability compared with that of the untreated control group (UCG) (*p* < 0.05). The cytokines TNF-α and TGF-β1 (5, 10, and 15 ng/mL) did not affect the viability of these cells. Therefore, we selected the concentrations of 1 mM of PTX and 10 ng/mL for TNF-α or TGF-β1 and proceeded to evaluate the viability of CaSki cells treated with PTX, TNF-α, and TGF-β1 and their combinations. In general, the treatments did not affect the viability of the CaSki cells (Figure 1B). In particular, we observed a decrease in the viability of the cells when the CaSki cells were treated with PTX + TNF-α + TGF-β1 (*p* < 0.05 in comparison with the UCG).

### 2.2. PTX Decreased TNF-α- and TGF-β1-Induced EMT in CaSki

The decrease of E-cadherin and increase of N-cadherin and Vimentin is the hallmark of EMT. For this reason, we assessed E-cadherin, N-cadherin, and Vimentin by Western blot assay in CaSki cells exposed to TNF-α or TGF-β1 or their combination and treated or not with PTX (Figure 2A). We observed that E-cadherin expression was significantly decreased in all treatment groups compared with the UCG (*p* < 0.05), as shown in Figure 2B. Likewise, TGF-β1 and TNF-α + TGF-β1 increased N-cadherin expression (*p* < 0.05) compared with the UCG and PTX groups; PTX decreased by 3.5-fold the N-cadherin expression induced by TNF-α + TGF-β1 (*p* < 0.05) (Figure 2C). TGF-β1 induced Vimentin expression (four-fold compared with the UCG; *p* < 0.001 (Figure 2D); Vimentin was markedly increased in the TNF-α + TGF-β1-treated group (10-fold compared to the UCG (*p* < 0.05). Additionally, we can observe in Figure 2D that PTX decreased by 6.1-fold the Vimentin expression induced by TNF-α plus TGF-β1 (*p* < 0.05). These results indicated that PTX was able to decrease TNF-α/TGF-β1-induced N-cadherin and Vimentin protein expression, which participate in the EMT process.

### 2.3. PTX Inhibited Cell Proliferation

CaSki cells were exposed to TNF-α (10 ng/mL) or TGF-β1 (10 ng/mL) alone or in combination and treated or not with PTX 1 mM for 5 days. We did not observe a significant increase in proliferation in cells treated with TNF-α or TGF-β1 alone or in combination (where it slightly decreased by 13% compared with the UCG) (Figure 3A). The results showed that cell proliferation was significantly decreased in all groups treated with PTX compared with the UCG (around 50%), reaching a 48% reduction in the PTX + TNF-α + TGF-β1 group (*p* < 0.05, compared with the UCG). We observed a significant decrease in the PTX-alone group (28% less proliferation than the UCG). Figure 3B presents representative images of CaSki cells treated with PTX and cytokines alone or in combination. These results suggest that PTX inhibited cell proliferation in CaSki cells.

### 2.4. PTX Decreased the Migration of CaSki Cells

The majority of cancer-related deaths among patients with solid tumors are caused by metastases, in which tumor cells migrate from the primary tumor to distant sites. CaSki cervical cancer cells were exposed to TNF-α and TGF-β1 and treated or not with PTX for 5 days, and a wound-healing assay was determined to examine the effect of PTX on cell migration by taking photographs at 0, 6, 12, and 24 h after wound generation. Figure 4 depicts the results of the migration assay at 12 h after the treatment. As shown in Figure 4A, no influence on cell migration was observed after treatment with TNF-α compared with the UCG. When the CaSki cells were exposed to TGF-β1, this resulted in a non-significant increase in migration (Figure 4B). In the case of the cells treated with TNF-α + TGF-β1, no immediate increase in migration was observed in comparison to the TGF-β1-treated group (Figure 4C); these results suggest that TGF-β1 increased CaSki-cell migration at 12 h. In contrast, cell migration was affected by PTX in all treated groups compared with the UCG. These decreases in migration continued until after 12 h. As shown in Appendix A, PTX significantly inhibited the migration activity of CaSki cells induced by TGF-β1 and TNF-α + TGF-β1 at 24 h after treatment. All of the results suggest that PTX prevented migration in mesenchymal-like CaSki cells.

### 2.5. PTX Decreased the Invasion of CaSki Cells

In order to evaluate whether PTX affects the migration of CaSki cells, we performed the invasion by transwell assay. Our findings indicated that TNF-α and TGF-β1 (Figure 5A,B) significantly increased the number of CaSki cells that passed through the transwell membrane compared with the UCG (*p* < 0.05). TNF-α + TGF-β1 significantly increased the number of cells passing through the membrane (*p* < 0.05). These results revealed that the interaction between cytokines could enhance the aggressive capacity in mesenchymal-like CaSki cells. Interestingly, we observed that PTX significantly decreased the number of cells passing through the membrane (*p* < 0.05). These results suggest that PTX can suppress CaSki-cell invasivity.

### 2.6. PTX Decreased NF-κB Activity

It is known that NF-κB also induces EMT in tumor cells [21]. To assess the impact of PTX on inhibiting NF-κB in tumor cells after 5 days, we first examined NF-κB activity in the epithelial CaSki cells (time 0 h). Figure 6A,B depict the phosphorylation of IκBα and p65 at 10 min, 30 min, and 1 h after treatment (Appendix A). TNF-α significantly increased IκBα and p65 phosphorylation (*p* < 0.05). Both PTX and PTX + TNF-α demonstrated a decrease in p65 phosphorylation. Following the addition of TGF-β1 or TNF-α + TGF-β1, the phosphorylation was not significantly increased. These results first indicated that NF-κB activity was present in epithelial CaSki cells and showed that PTX decreased the phosphorylation produced by TNF-α and TGF-β1. Next, we analyzed NF-κB activity in mesenchymal-like CaSki cells at 5 days (Figure 6C,D). IκBα and p65 phosphorylation levels were increased when TNF-α was added. However, the combination of TNF-α + TGF-β1 was more effective in increasing IκBα phosphorylation, the inhibitory subunit of NF-κB, suggesting that NF-κB was activated. These data reveal that TGF-β1 can increase NF-κB activity in mesenchymal-like CaSki cells. Moreover, we observed that PTX reduced IκBα and p65 phosphorylation levels. Collectively, we observed that TGF-β1 modulated the NF-κB pathway, which is involved in proliferation, invasion, and EMT. Our results suggest that PTX disrupts this pathway, which is essential in these protumor processes.

### 2.7. PTX Decreased SERPINE1 Gene Expression

We determined the possible effects of PTX in TGF-β1/Smad signaling in mesenchymal-like CaSki cells; thus, we analyzed *SERPINE1* gene expression in epithelial CaSki cells (time 0) after 4 h of treatment by qPCR (Figure 7A). The results showed that PTX alone did not decrease the expression of *SERPINE1*, while TNF-α did not significantly increase its expression. However, the TGF-β1 group indeed does, increasing its expression by about 64- and 55-fold more (with the reference genes *RPL32* and *RPLP0*, respectively) than the UCG (*p* < 0.05) and the TNF-α + TGF-β1 group increased its expression by about 12- and 1.7-fold more than the UCG (*p* < 0.05). Interestingly, the PTX treatment induced the downregulation of the *SERPINE1* gene in the group exposed to these cytokines (*p* < 0.05). These results suggested that TGF-β1 triggered the Smad pathway and that PTX decreased this activity. As shown in Figure 7B, PTX and TNF-α did not induce a significant change in the expression of the *SERPINE1* gene at 5 days. As expected, *SERPINE1* expression was significantly increased in cells treated with TGF-β1 (36- and 48-fold with *RPL32* and *RPLP0*, respectively) and TNF-α + TGF-β1 (105- and 71-fold with *RPL32* and *RPLP0,* respectively) compared with the UCG. As observed at 4 h, PTX significantly decreased *SERPINE1* expression in CaSki cells exposed to TGF-β1 (*RPL32* and *RPLP0*) at 5 days. Likewise, the same effect was observed in cells exposed to TNF-α + TGF-β1 treated with PTX, where PTX decreased 52.1- and 36.4-fold less (*RPL32* and *RPLP0*) in comparison with CaSki cells exposed only to TNF-α + TGF-β1 cytokines. This finding suggests that PTX decreased *SERPINE1* gene expression in mesenchymal-like CaSki cells.

## 3. Discussion

It has been reported that TNF-α and TGF-β1 play a crucial role in cancer initiation, proliferation, progression, metastasis, and EMT induction by facilitating the loss of epithelial proteins and increasing the upregulation of mesenchymal proteins [4,22]. The effects of TNF-α and TGF-β1 on EMT induction have not, to our knowledge, been fully investigated. In this study, we found that TNF-α and TGF-β1 in the cell model of advanced cervical cancer upregulated N-cadherin and Vimentin proteins, which contributed to the increased proliferation and invasion of CaSki cells. Crosstalk between TNF-α and TGF-β1 is associated with NF-κB and TGF-β1 signalings to promote EMT. It is well known that TNF-α and TGF-β1 can enhance the activation of NF-κB and TGF-β1 pathways. Here we report that the phosphorylation of IκBα and p65 and the upexpression of the *SERPINE1* positively affected EMT induction. The present work shows evidence that PTX decreased TNF-α/TGF-β1-induced EMT in cervical cancer cells via NF-κB and TGF-β1/Smad pathways. PTX counteracted TNF-α and TGF-β1 induced EMT in cervical cancer cells via NF-κB and TGF-β1/Smad pathways.

PTX is a methylxanthine derivative that inhibits phosphodiesterases [23]. Due to its anti-inflammatory effect, PTX is employed in a range of diseases and pathologies, including dermatological, hemorheological, cardiac, and COVID [24]. Its anti-inflammatory effects are due to the reduction in the production of proinflammatory cytokines such as TNF-α, IL-6, IL-1β, TGF-β1, and INF-γ by downregulating activation of the NF-κB and NFTA transcriptor factor [13,14,25]. PTX has been shown to increase apoptosis in several tumor cells and to sensitize these to antitumor drugs; in addition, it improves the prognosis of patients with hepatocarcinoma and significantly increases the apoptosis of leukemic cells in pediatric patients with ALL [11,26]. It has been reported that PTX in vivo and in vitro exerts antimetastatic and anti-angiogenic activities in B16F10 murine melanoma cells, in A375 melanoma cells, and in MDA-MB-231 breast cancer cells [27,28,29]. The authors suggested that PTX reduced the expression of αv α2, α3, α5, β1, β3, and β5 integrins and decreased the secretion of MMP-2 and MMP-9 [30] by suppressing the expression of the *PAI-1* gene and fibronectin [31]. Additionally, PTX is efficient in inhibiting TGF-β1-mediated collagen expression, blocking TGF-β1/Smad signaling in pulmonary fibrosis. PTX also blocks TGF-β1 expression and Smad2/3 activation in renal fibrosis and decreases *PAI-1/SERPINE1* expression in irradiated lung tissues and epithelial cells, thus preventing fibrosis [19,20,32]. Despite this, the effects of PTX on EMT have not, to our knowledge, been investigated.

In the present study, we applied molecular and cellular approaches to generate scientific data concerning the effects of PTX on mesenchymal cells. Results indicated that PTX might exert anti-EMT effects. The major finding of the present study was that PTX significantly decreased N-cadherin and Vimentin (Figure 2). To our knowledge, this is the first time that this effect of PTX has been reported.

On the other hand, under the same experimental conditions, we observed that cell migration was affected by PTX treatment principally at 24 h (Appendix A). However, it is reasonable to think that the results obtained from the wound-healing assay at 12 h differentiate migration from proliferation (Figure 4). In addition, the same behavior is observed in the cell invasion assay. PTX decreases the invasiveness capacity of tumor cells in a significant manner.

Interestingly, we observed that NF-κB and TGF-β1/Smad activities (indicated by the phosphorylation of IκBα and p65 proteins and the upexpression of the *SERPINE1* target gene, respectively) were upregulated by the TNF-α/TGF-β1 treatment. The NF-κB pathway is a crucial regulator of inflammation. Additionally, NF-κB signaling is essential for the induction and maintenance of EMT. Our findings revealed that PTX decreased NF-κB signaling and reduced EMT characteristics in CaSki cells. On the other hand, it is known that TGF-β1 facilities tumor invasion and metastasis [33]. In particular, TGF-β is responsible for reprogramming EMT genes. *PAI-1/SERPINE1* is the major physiological regulator of the plasmin-generating cascade and is a prominent member of the subset of the TGF-β-induced EMT-associated genes in malignant cells. *PAI-1/SERPINE1* expression facilitates tumor invasion through the control of proteolytic activity [34,35].

In our study, we found that TGF-β1 alone promoted the expression of *SERPINE1* and that the combination of TNF-α + TGF-β1 synergized *SERPINE1* expression. Additionally, the synergy between them contributed to the enhanced aggressive capacity of the cells. Intriguingly, we observed that TGF-β1 could cooperate with TNF-α to induce NF-κB activity for promoting EMT in CaSki cells. Indeed, it is currently thought that the activities of TNF-α and TGF-β1 arise as a result of the synergistic action of the NF-κB/TGF-β pathways; these findings provide evidence that NF-κB and TGF-β signaling enhances the EMT and the invasiveness of cervical cancer cells. Interestingly, we observed that the most significant effect of PTX occurred in the group treated with TNF-α + TGF-β1; these findings provide evidence that PTX can reduce the synergic effect between cytokines in EMT induction. These data would also provide a better understanding of the complex role of TNF-α and TGF-β1 in cervical cancer and support the idea that PTX decreases NF-κB and TGF-β1/Smad activity in mesenchymal-like CaSki cells.

In conclusion, our data demonstrate that PTX decreases the cell proliferation, migration, and invasion of mesenchymal-like CaSki cells. A key finding of this study is that PTX downregulates NF-κB signaling and *SERPINE1* expression and decreases the N-cadherin and Vimentin proteins. To our knowledge, this is the first time that this effect of PTX has been reported. PTX treatment may be a promising strategy for inhibiting EMT and treating cervical cancer and metastasis.

## 4. Materials and Methods

### 4.1. Culture Cells

Human cervical cancer CaSki cells (ATCC, Manassas, VA, USA #CRL-1550) were provided by Prof. Adriana Aguilar-Lemarroy. Cells were authenticated utilizing the Multiplex Cell Authentication system by Multiplexion GmbH (Friedrichshafen, Germany). The CaSki cells were cultured in Dulbecco’s Modified Eagle’s Medium (DMEM), low glucose, pyruvate (cat: 11885084; Gibco, Thermo Fisher Scientific, Waltham, MA, USA) supplemented with 10% (*v*/*v*) Fetal bovine serum (FBS) (cat: 26140079; Gibco, Thermo Fisher Scientific, Waltham, MA, USA), 100 mg/mL Streptomycin, and 100 units/mL Penicillin (cat: 15140122; Gibco; Thermo Fisher Scientific, Waltham, MA, USA) in a humidified incubator with 5% CO_2_ and a 95% air atmosphere at 37 °C. The cells were passaged once they reached 75–85% confluence. Prior to the initiation of all experiments, cell viability was determined with Trypan Blue (Sigma-Aldrich, St. Louis, MO, USA) (viability > 95%). The cell line was tested for mycoplasma contamination employing the Universal Mycoplasma Detection Kit (ATCC, Manassas, VA, USA); the cells were negative throughout the study. When CaSki cells reached 85% confluence, they were harvested with 0.25% Trypsin-EDTA (cat: 25200-072; Gibco, Thermo Fisher Scientific, Waltham, MA, USA) and seeded at different cell densities according to each subsequent experiment.

### 4.2. Drugs and Reagents

PTX (cat: P1784) was obtained from Sigma-Aldrich (St. Louis, MO, USA). Recombinant Human TGF-β1 (cat: 580704) and recombinant Human TNF-α (570104) were obtained from Biolegend (San Diego, CA, USA). All stock solutions were diluted in culture media prior to use.

### 4.3. Induction of EMT Cells

CaSki cells were seeded at a specific density (2 × 10^4^ cells/mL) and cultured overnight for their attachment. The culture medium was replaced, and the cells were treated with TGF-β1 (10 ng/mL) and TNF-α (10 ng/mL) either alone or in combination for 5 days. Cultures were restimulated every 2.5 days. Untreated control cells were cells cultured at the same time. E-cadherin, Vimentin, and N-cadherin protein expression were analyzed at 5 days.

### 4.4. Experimental Conditions

CaSki cells were treated for 5 days with PTX (1 mM), TNF-α (10 ng/mL), and TGF-β1 (10 ng/mL), either alone or in their simultaneous combinations. The addition of PTX was performed 1 h prior to treatment. Viability, E-cadherin, N-cadherin, Vimentin, proliferation, migration, invasion, and *SERPINE1* gene expression were assessed at 4 h and 5 days after treatment exposure. ICW assays were performed to detect NF-κB activity at 1 h (time 0) and 5 days after treatment.

### 4.5. Cell Viability by 7-AAD Staining for Flow Cytometry

CaSki cells (1 × 10^5^ cells) were seeded on 24-well plates and treated with the following: PTX (0.5, 1, 2, and 4 mM); TNF-α (5, 10, and 15 ng/mL); TGF-β1 (5, 10, and 15 ng/mL); PTX (1 mM) + TNF-α (10 ng/mL); PTX + TGF-β1 (10 ng/mL); TNF-α (10 ng/mL) + TGF-β1 (10 ng/mL), and PTX (1 mM) + TNF-α (10 ng/mL) + TGF-β1 (10 ng/mL) for 5 days. After the cells were harvested, two washes were carried out with Phosphate buffered saline solution (PBS, cat: P4417-100TAB, Gibco™; Thermo Fisher Scientific, Inc., Waltham, MA, USA), and the cells were stained with 7AAD (cat: 420403 BioLegend San Diego, CA, USA) for viability detection. Finally, 10,000 events were acquired through the Attune Acoustic Focusing Cytometer (Applied Biosystems, Waltham, MA, USA). The data were analyzed with FlowJo X 10.0.7r2 statistical software (BD Biosciences, Franklin Lakes, NJ, USA). Data are shown as the mean ± SD represented as the percentage of viability.

### 4.6. SRB Proliferation Assay

The SRB assay was performed as described by Skehan et al. [36]. Briefly, CaSki cells were seeded on 96-well plates at a density of 1 × 10^4^ cells in 200 µL per well. After 5 days of exposure to PTX (1 mM), TNF-α (10 ng/mL), and TGF-β1 (10 ng/mL) either alone or in combination, and then cells were fixed for 10 min in 4% paraformaldehyde solution in PBS, then were incubated for 15 min, and were subsequently incubated for 5 min with SRB solution at 0.4%. Then the cells were washed with acetic acid (1%) and analyzed under a Primover inverted phase contrast microscope (ZEISS, Oberkochen, Germany). Subsequently, the SRB was solubilized. Absorbance at 510 nm was measured with a Synergy HT Multi-Mode Micro Plate Reader (Biotek, Winooski, VT, USA). Data are presented as the mean ± SD represented as the percentage of proliferation.

### 4.7. Western Blot

CaSki (3 × 10^6^) were seeded on p150 Petri dishes and cultured at 37 °C in culture medium for 24 h. Then, the cells were washed with PBS and cultured for 5 days at 37 °C, with 1 mM PTX and with TNF-α and TGF-β1 (10 ng/mL each) either alone or in combination. We evaluated the expression of E-cadherin, N-cadherin, and Vimentin in cervical-cancer cells. In brief, the cells were harvested and lysed with 300 µL of RIPA buffer (0.5% deoxycholate, 0.5% NP-40, 0.5% SDS, 50 mM Tris pH 8.0, and 150 mM NaCl) with a protease inhibitor cocktail (cat: 1183617001; Roche Applied Science, Penzberg, DE, USA) and incubated on ice for 30 min. Next, the lysate was passed through a blunt 20-G needle-fitted syringe 30 times and was sonicated (5 min, high levels, with a 30-s on–off time interval) utilizing the Sonicator bioruptor (cat: B01060010; Diagenode, Liège, Belgium). The cells were incubated for 30 min, and then the solution was centrifuged (5 min at 12,000 rpm at 4 °C) for protein collection; the proteins were quantified with the Bradford Assay kit (Bio-Rad, Hercules, CA, USA), and primary electrophoresis was performed with polyacrylamide gels employing 50 µg of total protein. Next, the proteins were transferred onto a PVDF membrane (0.2-µm; cat: ISEQ 00010; Immobilon-PSQ) through a semidry system (Bio-Rad, Hercules, CA, USA), blocked for 2 h with 1X Western Blocking Reagent (LI-COR Biosciences, Lincoln, NE, USA), and incubated under agitation overnight with a primary antibody (1:1000 at 4 °C. The following day, these were incubated with immunofluorescence IRDye 800CW anti-goat IgG secondary antibody (1:15,000) for 2 h at room temperature. The membranes were washed and revealed using the Odyssey™ Infrared Imaging System (LI-COR Biotechnology, Lincoln, NE, USA). The primary antibodies employed in the assays were anti-E-cadherin (1:400, sc-8426, clone no: G-10; Santa Cruz Biotechnology, Inc., Dallas, TX, USA), anti-Vimentin (1:400, sc-6260, clone no: V9; Santa Cruz Biotechnology, Inc., Dallas, TX, USA) and anti-N-cadherin (1:400, sc-59987, clone no: 13A9; Santa Cruz Biotechnology, Inc., Dallas, TX, USA). Optical density (OD) was measured utilizing Image Studio Lite ver. 5.2.5 software and normalized to Revert™ 700 Total Protein Stain (cat: P/N 926-11010; LI-COR, Biosciences, Lincoln, NE, USA) according to the manufacturer’s instructions.

### 4.8. In Vitro Migration Assays

A scratch wound-healing assay was performed to determine cell migration. For this, cells were seeded at a density of 1 × 10^5^ cells/mL on a 6-well tissue-culture plate, allowing for overnight attachment. The following day, the cells were treated with 1 mM PTX, TNF-α, and TGF-β1 (10 ng/mL each) either alone or in combination for 5 days. For the wound-healing assay, the monolayers were vertically scratched using a sterile p200 pipette tip after the cells reached a confluence of 90–95%, and these were later washed to eliminate the detached cells. A control photographic image was taken using the AxioCam ERc5s inverted phase contrast microscope (Primo Vert, cat. no. 415510-1101-000; Zeiss AG) at 40× magnification. The rate of cell migration was measured as the percentage of wound area occupied by the cells compared with the initial wound area using ImageJ software (version 1.8.0_172; National Institutes of Health). Wound healing (%) was calculated as follows:(1)$$Wound\;healing(\%)=(\frac{{At}_{0}-{At}_{t}}{{At}_{0}})\times 100$$
where:*At*_0_ = is the area of the wound measured immediately after scratching (*t*_0_ = area at time 0)*At_t_* = is the area of the wound measured h hours after the scratch is performed.

### 4.9. Cell Invasion Assay

CaSki cells (1 × 10^5^) were seeded on a 6-well tissue-culture plate, and 24 h after cell culture, they were treated with 1 mM PTX and TNF-α and TGF-β1 (10 ng/mL each) either alone or in combination for 5 days. Then the migration of the CaSki cells was measured by chemotactic directional migration by using a 24-well Transwell insert. The 8-μm pore filters (Wuxi NEST Biotechnology Co., Ltd., Wuxi, China) were coated with 0.25 mg/mL Matrigel for 24 h, and the CaSki cells (1.5 × 10^5^ cells/0.15 mL) were placed in the upper chamber and allowed to undergo migration for 24 h. The non-migrated cells in the upper chamber were removed with a cotton swab. The filters were stained with SRB 4%. The migrated cell chambers that adhered to the underside of the filter were counted and photographed using a Zeiss Primo Vert microscope (cat: 415510-1101-000; ZEISS) at 100×. The SRB was solubilized. Absorbance at 510 nm was measured with a Synergy HT MultiMode Micro Plate Reader (Biotek, Winooski, VT, USA). Data are shown as the mean ± SD represented as the invasion. Invasion (%) was normalized to the untreated control as follows:(2)$$Invasion(\%)=(\frac{ab\;group\;T}{ab\;group\;UCG}\times 100)$$
where:*ab group T* = absorption of the treated group*ab group UCG* = absorption of the untreated group

### 4.10. ICW Assay

The CaSki epithelial cervical cancer cells (9 × 10^4^ cells/wells) and mesenchymal-like CaSki cells (3.5 × 10^4^ cells/wells) were grown in a 96-well optical black-walled transparent-bottom plate (Thermo Fisher Scientific, Waltham, MA, USA). The cells were allowed to attach to the plastic and then were treated with 1 mM PTX and TNF-α and TGF-β1 (10 ng/mL each) alone or in combination for 60 min. Then the cells were fixed using 100 mL of methanol-acetone solution (3:1) for 20 min at −21 °C, and cell permeabilization was performed twice with 0.5% Triton-X-100 for 5 min at room temperature; then, the cells were incubated at 4 °C overnight. The following day, the cells were blocked with LI-COR Odyssey blocking solution (LI-COR Biosciences, Lincoln, NE, USA) for 4 h and incubated with the primary mouse IgG antibodies against anti-IκBα (1:400 dilution; Cat. no. L35A5; Cell Signaling Technology Inc., Danvers, MA, USA), anti-NF-κB-p65 (1:400 dilution; Cat. no. L8F6; Cell Signaling Technology Inc.), anti-Phospho-IκBα (Ser32; 1:400 dilution; Cat. no. 14D4; Cell Signaling Technology Inc.), and anti-Phospho-NF-κB-p65 (Ser536; 1:400 dilution; Cat. no. 93H1; Cell signaling Technology Inc.) for 72 h. Finally, the cells were washed and incubated with the goat anti-mouse IgG IRDye™ 800 secondary antibody (dilution 1:15,000; LI-COR Biosciences, Lincoln, NE, USA) for 2 h at room temperature. The protein expression obtained by means of the integrated fluorescence intensities was detected using the Odyssey CLx Team ICW station (Odyssey Software version 3.0; LI-COR Biosciences, Lincoln, NE, USA). Analysis of the relative expression of the IκBα and p65 total and phosphorylated were performed by normalization with DRAQ5 staining (LI-COR Biosciences, Lincoln, NE, USA). Data are presented as the ratio of phosphorylation as follows:(3)$$Ratio\left(\frac{Phospho}{Total}\right)=\frac{NSPP}{NSTP}$$
where:*NSPP:* Normalized signal phosphor protein*NSTP:* Normalized signal total protein

### 4.11. Reverse Transcription-(RTq)-PCR

To determine the mRNA expression of the *SERPINE 1* gene, we proceeded to perform the RTq-PCR methodology. Briefly, 3 × 10^6^ cells were seeded on p100 Petri dishes (cat: 353003; Costar) and treated for 4 h or 5 days with the indicated treatments. Then mRNA extraction was carried out according to the specifications of the RNeasy Plus Mini Kit (cat: 74134; Qiagen, Hilden, Germany); 970 ng of total RNA was used for reverse transcription using the SuperScript™ III First-Strand Synthesis SuperMix kit (cat: 04896866001; ROCHE). Then, 2 µL of cDNA was amplified with the LightCycler FastStart DNA Master PLUS SYBR Green I kit (cat: 18080-400). qPCR was performed using the LightCycler ver. 2.0 Instrument (both Roche Diagnostic GmbH). Data were normalized via the 2^−ΔCq^ method, using *RPL32* and *RPLP0* as reference genes. The primer sequences were as follows: *SERPINE1* (NC_000007.14), sense 5′ ATT CTG AGT GCC CAG CTC ATC 3′, antisense 5′ CTC GTG AAG TCA GCC TGA AAC 3′; *RPL32* (NC_000003.12)*,* sense 5′ GCA TTG ACA ACA GGG TTC GTA 3′ antisense 5′ ATT TAA ACA GAA AACGTG CACA 3′, and *RPLP0* (NC_000012.12), sense 5′ CCT CAT ATC CGG GGG AAT GTG 3′, antisense 5′ GCA GCA GCT GGC ACC TTA TTG 3′.

### 4.12. Statistical Analysis

Normality, homogeneity of variance, and data independence were determined before each analysis. Statistical analyses were performed with GraphPad Prism ver. 8.1 statistical software (San Diego, CA, USA), and the latter was employed for all analyses. One-way ANOVA followed by the Tukey pos-hoc test were employed to compare groups. For assay results, we applied the Mann–Whitney *U* test; *p* < 0.05 indicates statistical differences.

## 5. Conclusions

In conclusion, our studies, to our knowledge for the first time, provided evidence of the antimetastatic activity of PTX in counteracting TNF-α/TGF-β1-induced EMT in cervical cancer cells via the NF-κB/TGF-β1/*SERPINE1*.

## Figures and Tables

**Figure 1 ijms-24-10592-f001:**
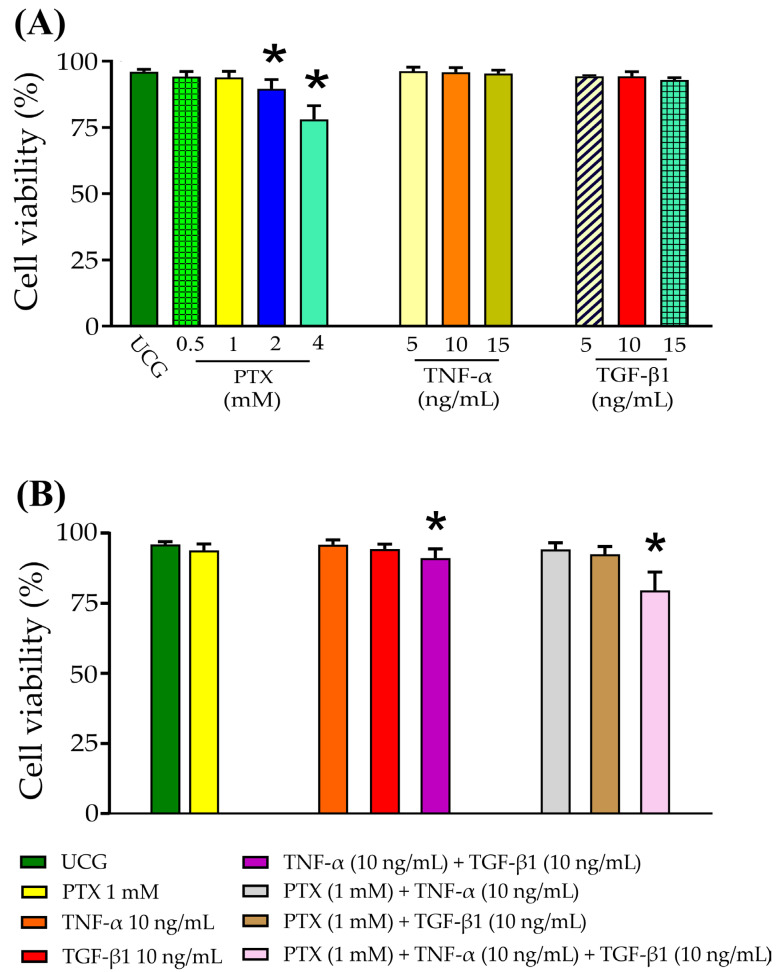
Assessment of the viability of CaSki cells treated with PTX, TNF-α, and TGF-β1 by flow cytometry. (**A**) CaSki cells were treated for 5 days with different doses of PTX (0.5, 1, 2, and 4 mM) and TNF-α or TGF-β1 (5, 10, and 15 ng/mL); viability was determined using 7AAD by flow cytometry. (**B**) Cervical cancer CaSki cells were treated with PTX (1 mM), TNF-α (10 ng/mL), and TGF-β1 (10 ng/mL) alone or in combination for 5 days, and viability was determined by flow cytometry. Data are represented as mean ± standard deviation (SD) of three independent experiments performed in triplicate. Statistical analysis: Mann-Whitney *U* test * *p* < 0.05 = statistical significance compared with the UCG.

**Figure 2 ijms-24-10592-f002:**
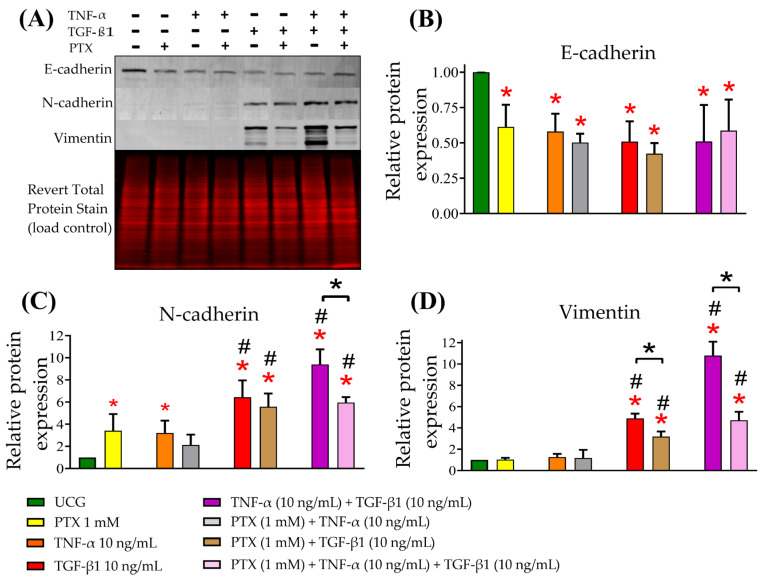
Expression of EMT-associated markers: E-cadherin, N-cadherin, and Vimentin in CaSki cells exposed to TNF-α, TGF-β1, and their combination and treated or not with PTX. (**A**) Western blotting was utilized to evaluate the differences in E-cadherin, N-cadherin, and Vimentin expression. (**B**) Densitometry analyses of E-cadherin. (**C**) Densitometry analyses of N-cadherin in CaSki cells cultured in the presence of TGF-β1 and TNF-α, treated or not with PTX. (**D**) Densitometry analyses of Vimentin in CaSki cells cultured in the presence of TGF-β1 and TNF-α, treated or not with PTX. Revert Total Protein Stain was used for normalizing data. Data are represented as mean ± SD of three independent experiments performed in triplicate. Statistical analysis: Mann-Whitney *U* test. *
*p* < 0.05 = statistical significance compared with the UCG. # *p* < 0.05 statistical significance in comparison with PTX groups, and * *p* < 0.05 statistical significance in the comparison between groups.

**Figure 3 ijms-24-10592-f003:**
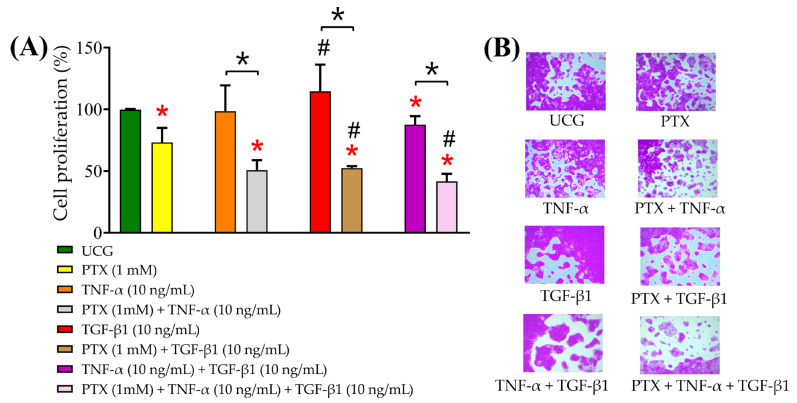
Determination of CaSki cell proliferation exposed to TNF-α, TGF-β1, and their combination treated with or without PTX for 5 days. (**A**) The cells were treated with PTX, TNF-α, or TGF-β1 alone or their combination for 5 days; after that, proliferation was determined using the Sulforhodamine B (SRB) assay, the scale bars denote 100 µm (40× amplification). (**B**) Representative images of cells stained with SRB in each treatment group (40× amplification). Data are represented as mean ± SD from three independent experiments conducted in triplicate. Statistical analysis: Mann-Whitney *U* test. *
*p* < 0.05 statistical significance in comparison with the UCG, # *p* < 0.05 statistical significance in comparison with PTX groups, and * *p* < 0.05 statistical significance in the comparison between groups.

**Figure 4 ijms-24-10592-f004:**
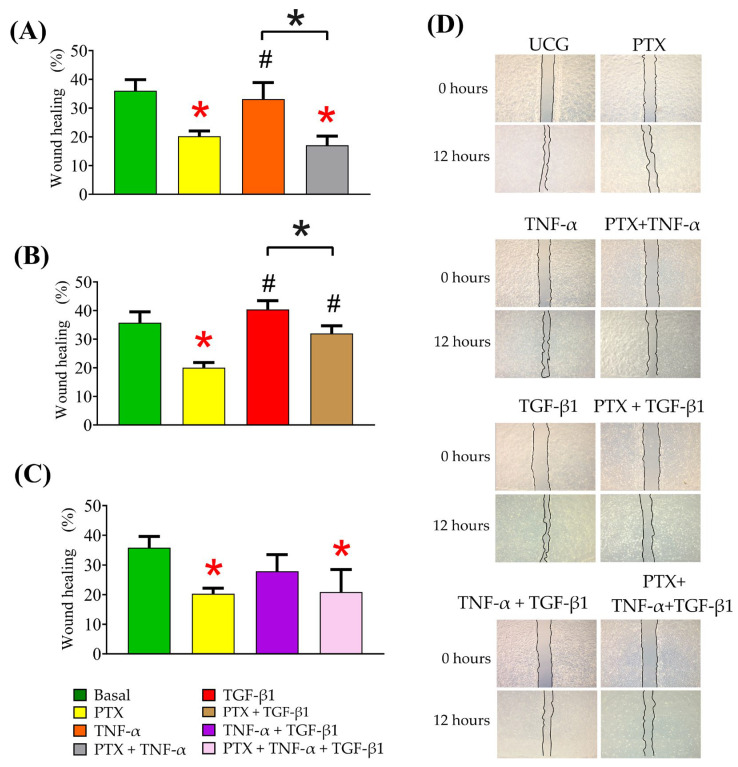
Determination of CaSki cell migration exposed to TNF-α, TGF-β1 and their combination treated with or without PTX for 5 days and subjected to the scratch wound-healing assay over 12 h. (**A**) Effect of TNF-α on the migration of CaSki cells. (**B**) Effect of TGF-β1 on migration. (**C**) Effect of TNF-α + TGF-β1 on the migration of CaSki cells. (**D**) Representative images from a wound-healing assay taken at 12 h, the scale bars denote 100 µm (40× amplification). White dotted lines indicate wound margins. Data are represented as the mean ± SD of three independent experiments performed in triplicate. Statistical analysis: Mann-Whitney *U* test *
*p* < 0.05 statistical significance in comparison with the UCG. # *p* < 0.05 statistical significance in comparison with PTX groups, and * *p* < 0.05 statistical significance in the comparison between groups.

**Figure 5 ijms-24-10592-f005:**
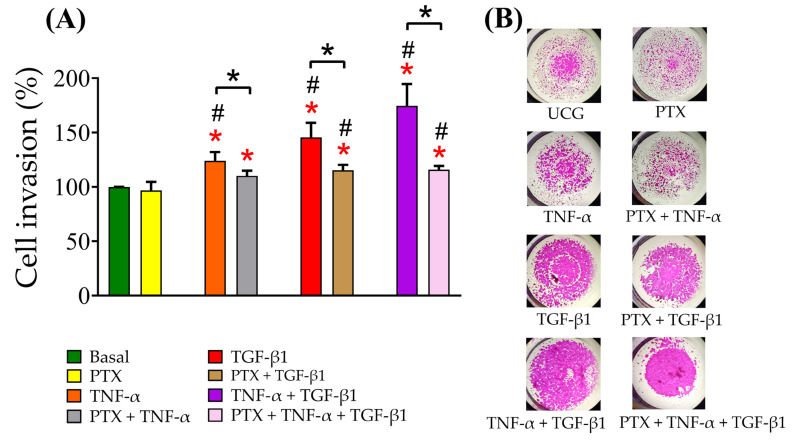
Determination of CaSki cell invasion exposed to TNF-α, TGF-β1 and their combination treated with or without PTX for 5 days and subjected to the chemotaxis assay over 24 h. (**A**) Effect of PTX on invasive capacity induced by TNF-α, TGF-β1 alone or in combination in the CaSki cells. (**B**) A chemotaxis assay was performed to evaluate invasive cells; representative images, the scale bars denote 100 µm (40× amplification). Data are represented as mean ± SD from three independent experiments, each run conducted in triplicate. Data are represented as the mean ± SD of three independent experiments performed in triplicate, normalized to the respective UCG. Statistical analysis: Mann-Whitney *U* test. *
*p* < 0.05 statistical significance in comparison with the UCG, # *p* < 0.05 statistical significance in comparison with PTX groups, and * *p* < 0.05 statistical significance in the comparison between groups.

**Figure 6 ijms-24-10592-f006:**
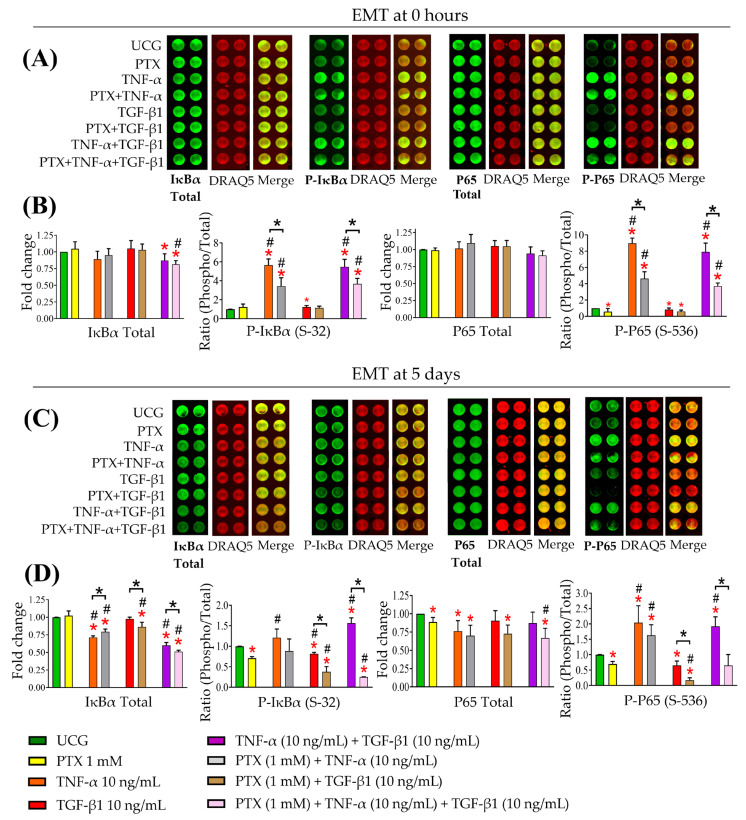
Determination of IκBα (S-32) and p65 (S536) phosphorylation in epithelial and mesenchymal-like CaSki cells exposed to TNF-α, TGF-β1 and their combination treated with or without PTX. The CaSki cells were exposed to these cytokines and treated or not with PTX for 10 min in time 0 and for 5 days. (**A**) Representative images of an In-Cell Western (ICW) assay plate in time 0. The proteins targeted were IκBα, phospho-IκBα, p65, and phospho-p65, examined in the epithelial CaSki cells (Time 0 h). (**B**) Ratio of IκBα total/phospho-IκBα and ratio p65 total/phospho p65 at 10 min. (**C**) Representative images of an ICW plate. The proteins targeted were IκBα, phospho-IκBα, p65, and phospho-p65, examined in the mesenchymal-like CaSki cells (5 days). (**D**) Ratio of IκBα total/phospho-IκBα and ratio p65 total/phospho p65 at 10 min. Data are represented as mean ± SD from three independent experiments performed in triplicate. Statistical analysis: Mann-Whitney *U* test. *
*p* < 0.05 statistical significance in comparison with the UCG, # *p* < 0.05 statistical significance in comparison with PTX groups, and * *p* < 0.05 statistical significance in the comparison between groups.

**Figure 7 ijms-24-10592-f007:**
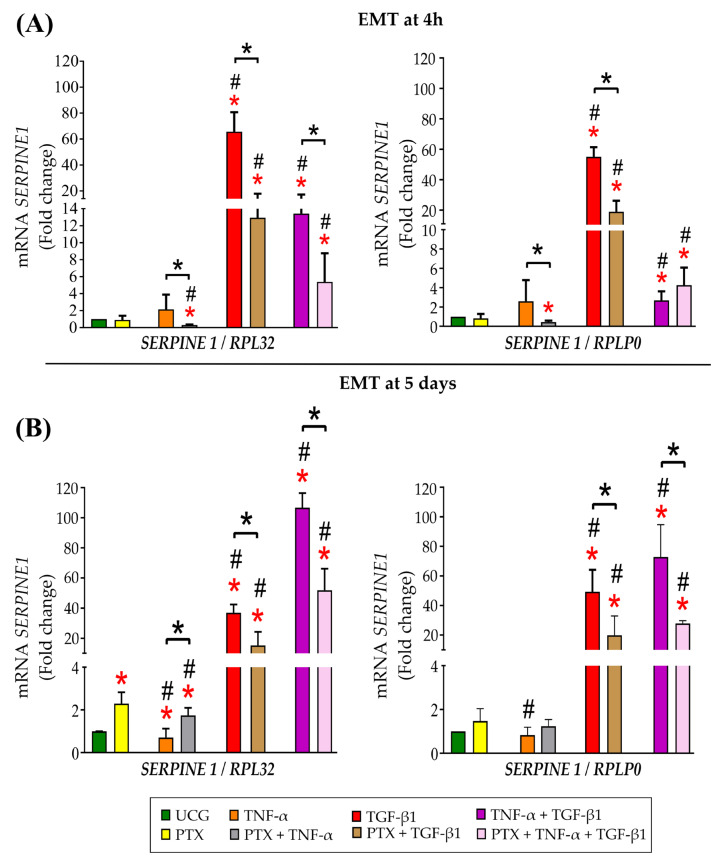
PTX downregulated *SERPINE1* expression in epithelial and mesenchymal-like CaSki cells exposed to TNF-α and TGF-β1 alone or in combination. (**A**) Epithelial CaSki cells were exposed to TNF-α or TGF-β1 or their combination and treated or not with PTX for 4 h; after that, we evaluated the expression of the *SERPINE1* gene. (**B**) CaSki cells exposed to TNF-α or TGF-β1 or their combination and treated or not with PTX for 5 days; we evaluated the expression of the *SERPINE1* gene. *SERPINE1* expression was evaluated by qPCR. *RPL32* and *RPLP0* were used as reference genes. Data are represented as mean ± SD from three independent experiments performed in triplicate. Statistical analysis: Mann-Whitney *U* test. *
*p* < 0.05 statistical significance in comparison with the UCG, # *p* < 0.05 statistical significance in comparison with PTX groups, and * *p* < 0.05 statistical significance in the comparison between groups.

## Data Availability

The data that support the findings of this study are available from the corresponding author upon reasonable request.

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
