# Peer review of "Pentoxifylline Inhibits TNF-α/TGF-β1-Induced Epithelial-Mesenchymal Transition via Suppressing the NF-κB Pathway and SERPINE1 Expression in CaSki Cells"

_ijms, 2023, doi:10.3390/ijms241310592_

Round 1
Reviewer 1 Report (Previous Reviewer 3)
This is a much better paper though I still believe you cannot make definitive statements when only one cell line is used. The fact that many papers only use one cell line is not any sort of justification, top quality journals do not tolerate such shortcomings. There are some minor errors but one that stands out is the fact that the authors continually refer to the CaSki cells as ‘mesenchymal cells’ when they have undergone some degree of EMT (e.g. lines 202, 230, 234, 244, 370). They have NOT changed their embryological origin, they are still epithelial cells; more correct to say ‘mesenchymal-like’ or ‘CaSki tumour cells with a more mesenchymal phenotype’
Minor points
- Line 49: ‘cell b decreasing…’
- Line 138: ‘stpatistical’
- Authors use ‘both CaSki’ and ‘CasKi’
- Line 220: ‘to induces..’ should be ‘..also induces..’
- Line 337: ‘is 0.7 times in 24h.’ This does not make sense
Only occasional errors, otherwise good
Author Response
Response to Reviewer 1 Comments
This is a much better paper though I still believe you cannot make definitive statements when only one cell line is used. The fact that many papers only use one cell line is not any sort of justification, top quality journals do not tolerate such shortcomings.
Point 1. There are some minor errors but one that stands out is the fact that the authors continually refer to the CaSki cells as ‘mesenchymal cells’ when they have undergone some degree of EMT (e.g. lines 202, 230, 234, 244, 370). They have NOT changed their embryological origin, they are still epithelial cells; more correct to say ‘mesenchymal-like’ or ‘CaSki tumour cells with a more mesenchymal phenotype’
Response 1. We have made these changes in response to your suggestion; now we use the term “mesenchymal-like”
Points 2. Minor points
Line 49: ‘cell b decreasing…’
This is corrected
Line 138: ‘stpatistical’
This is corrected
Authors use ‘both CaSki’ and ‘CasKi’
This is corrected
Line 220: ‘to induces..’ should be ‘..also induces..’
This is corrected
Line 337: ‘is 0.7 times in 24h.’ This does not make sense
This is eliminated effectively, this does not make sense in the discussion.
Comments on the Quality of English Language. Only occasional errors, otherwise good.
Thank you for taking the time to give us your valuable feedback. Your comments were extremely helpful and have significantly improved our work.

Reviewer 2 Report (Previous Reviewer 2)
The authors convinced me and I hereby support the publication.
Author Response
Response to Reviewer 2 Comments
Point 1 The authors convinced me and I hereby support the publication.
Thank you for taking the time to provide me with detailed and constructive feedback. Your comments were extremely helpful and have significantly improved our work.

Reviewer 3 Report (Previous Reviewer 1)
Please see attached report for detailed comments.

The writing in general is clear enough to convey the message.
Author Response
Response to Reviewer 3 Comments
In this revised submission, the authors present a fit-for-purpose design of the studies to test the hypothesis and a logical flow of analysis on the ques�ons to support their conclusions. I have some specific comments for this round of submission.
Thank you for taking the time to give us your valuable feedback. Your comments were extremely helpful and have significantly improved our work.
Point 1. On line 59, EMT needs to be fully spelled out as the first time appearing in the main context.
It is corrected
Point 2. For section 2.2, please add brief introductions of how the E-cadherin, N-cadherin, and Vimentin are related to EMT so that the readers can learn the biological significance and the reasoning of the conclusion of the analysis in this section.
We have added a brief introduction to section 2.2
Point 3. For section 2.4, since the 4 time points were mentioned in the context, I feel Figure S1A is a beter main figure to show than Figure 4A as it is clear and more informative.
We chose to show an intermediate point (12h) as an interesting result in the migration test. However, considering your suggestion, we have described the findings at 24 h in the results section.
Point 4. On line 335-336, 24h is not mentioned for Figure 3. Please check and make it clear.
It is corrected
Point 5. Describe/mention the number of replicates used and the error bars for all the bar plot comparisons.
It is corrected

This manuscript is a resubmission of an earlier submission. The following is a list of the peer review reports and author responses from that submission.
Round 1
Reviewer 1 Report
Please see attached for detailed comments.

Author Response
Thank you for taking the time to provide me with detailed and constructive feedback. Your comments were extremely helpful and have significantly improved our document.
Major concern1.
1.- Section 2.7 has been revised and rewritten.
2.- According to Figure 7, PTX treatment does not reduce the expression level of SERPINE1 in general (it increases the level in 2 out of 4 cases). The authors need to explore and present more clearly why the gene expression of SERPINE1 increased with PTX and PTX+TNF-a treatment but decreased with the other 2 treatment
Thank you for your observation,
After normalizing results obtained by qPCR of the PTX, PTX+TNF treated groups with the references genes, the increase of SERPINE1 was only significant when results were normalized with the RLP32; these may be due to the variability expression of the reference gene. In the PTX+TNF group, PTX does not increase SERPINE1 expression due to the combination; only the effect of PTX in this group is observed; there was no synergy between compounds to increase SERPINA1. Unfortunately, this finding is difficult to interpret because there is nothing in the literature about it. The effect of PTX on SERPINE1 expression was only noticeable after 5 days of treatment. This result could be explored further in another study. Nonetheless, the primary discovery is that PTX obstructs the effect of TGF-β1 on serpin after 4 hours and 5 days.
However, this result is significant in at least two specific findings.
First, the effect of PTX on TGB-β1 in the induction of SERPINA1 decreases its expression.
2cond. The effect of PTX on the action combined cytokines on the expression of SERPINA1.
Further studies will need to be undertaken.
The conclusion on lines 270-272 is an over-extrapolation of the experiment results.
The conclusions of lines 270-272 have been rewritten.
The authors need to present how exactly PTX inhibits the activation of which signaling pathways so as to make such conclusions. Related to the point above, the current title of the manuscript is not supported by the manuscript regarding its ‘suppressing NF-kB pathway and SERPINE1 expression’. Besides SERPINE1, a comparison analysis between EMT at 0 days and 5 days may need to be conducted to better support the title.
We have included information in the discussion regarding the signaling pathways that are impacted by pentoxifylline.
The results of the ICW analysis indicated a decrease in the phosphorylation of IκBα and p65 at both time 0 and after 5 days of treatment. Additionally, new results from the SERPINA1 expression at 0 and 5 days have been included, demonstrating that PTX alone does not increase SERPINA. However, when combined with all groups, PTX significantly decreases SERPINA. This effect of PTX is also observed at the 5-day mark. The results presented in our document are in line with its title.
The current version of the abstract does not accurately match and emphasize the title. Please revise.
We have made improvements and corrections to the abstract.
It is recommended to label the three bar plots in Figure 2 individually as B, C and D and revise section 2.2 by adding reference to Figure 2B, 2C and 2D when stating the corresponding observations.
This is corrected.
The current 2.2 is confusing as the readers do not which bar plot to look at since they are all referred as Figure 2B. 2.
This is corrected.
The red star is uncolored in Figure 2’s description.
This is corrected.
The statement on line 146-147 is unsupported by Figure 3. Please revise
This is corrected.
Add legends to the bar plots in Figure 6
This is corrected
The definition of the error bars and number of samples are missing in some of the figures. 6
This is corrected.
Please add figure legends and descriptions to all the supplementary figures.
Revised and added.
Supplementary figures 2, 4, and 5 are unmentioned throughout the manuscript.
The document has been updated, and now supplementary figures are cited in the document.
The English writing of the manuscript needs further checking and major revision.
A speaker of the English language has reviewed the document.

Reviewer 2 Report
In the study entitled "Pentoxifylline inhibits TNF-α/TGF-β1-induced epithelial-mesenchymal transition via suppressing NF-κB pathway and SERPINE1 expression in CaSki cells." by Palafox-Mariscal et al., the authors investigated the capacity of the Pentoxifylline to inhibit EMT induced by two inflammatory cytokines, TNFa and TGFb1 of a cervical cancer cell line. EMT is a critical process responsible for cancer cell metastatsis and identifying drugs that can prevent it is absolutely crucial to improve the current therapeutic strategies. Thus, the paper could be of interest of researchers <working in the filed. Nevertheless, the paper is very poorly written making most of the time very difficult to understand the point the authors want to make. The authors must perform an extensive english review of the text and, if possible, make it revised by a native english speaker.
I have also some major comments that need to be addressed before considering the current study suitable for publication.
1) The authors performed parametric Student t-test to test the significance of the difference they observed from three independent biological replicate. This is not the appropriate statistical test to make from such a small number of replicates. The authors should perform Mann-Whitney test and most likely additional replicates to validate the differences in the different conditions.
2) The authors claim that Pentoxifylline inhibits EMT induced by TGFb1 and TNFa. However, in most of the parameters they tested, the effect is already observed without the presence of the cytokines and the effect mediated by the cytokines mentioned by the authors is at least barely detectable. For example, in Figure 4, the authors claim that Pentoxifylline blocks cell cell migration induced by TNFa/TGFb1. If we look at the graphs, the effect of Pentoxifylline on wound healing is already observed without the cytokines. Besides, TNFa is not inducing cell migration at all and the effect of TGFb1 is barely detectable. So, what is really the effect of Pentoxifylline? Is it really blocking EMT or are the effects of Pentoxifylline mediated by a general effect on cell homeostasis?
3) As mentioned in comment 2, the authors claim that Pentoxifylline inhibits cell migration induced by TGFb1 and TNFa. Given the effects Pentoxifylline has on cell proliferation (Figure 3), the phenotype observed in Figure 4 could be due to the impairment of cell proliferation. Thus, further experiments are required to really discriminate between cell proliferation and migration.
Author Response
Thank you for taking the time to provide me with detailed and constructive feedback. Your comments were extremely helpful and have significantly improved our work.
Major concerns.
- 1) The authors performed parametric Student t-test to test the significance of the difference they observed from three independent biological replicate. This is not the appropriate statistical test to make from such a small number of replicates. The authors should perform Mann-Whitney test and most likely additional replicates to validate the differences in the different conditions.
The statistical analysis was changed and now our results show the mean ± standard deviation of three independent experiments, each performed in triplicate. An analysis Statistical with the Mann-Whitney test.
- 2) The authors claim that Pentoxifylline inhibits EMT induced by TGFb1 and TNFa. However, in most of the parameters they tested, the effect is already observed without the presence of the cytokines and the effect mediated by the cytokines mentioned by the authors is at least barely detectable. For example, in Figure 4, the authors claim that Pentoxifylline blocks cell migration induced by TNFa/TGFb1. If we look at the graphs, the effect of Pentoxifylline on wound healing is already observed without the cytokines. Besides, TNFa is not inducing cell migration at all and the effect of TGFb1 is barely detectable. So, what is really the effect of Pentoxifylline? Is it blocking EMT or are the effects of Pentoxifylline mediated by a general effect on cell homeostasis?
-Migration and invasion are functional assays to assess cellular capacities.
The migration tests provide data on spontaneous migration or response to a chemo-attractant; this method helps determine the migration ability of whole cell masses. The invasion test analyzes the ability of single cells to respond directionally to a chemo-attractant and migrate through a physical barrier toward it. Transwell membrane with Matrigel to mimic the process of extracellular matrix invasion and extravasation. The data obtained show: migration velocity and invasion capabilities. These assays do not show the loss of epithelial markers and the gain of mesenchymal markers. Our result did not show significant results between TGF-β1 alone or combined with TNF-α compared to untreated. Based on the data, there are two potential explanations to consider.
The first possibility is that the antiproliferative impact of PTX could impact migration.
The second possibility is that TNF-α and TGF-β1 have not stimulated migration.
- Indeed, the TNF-α treated group of cells did not increase migration and vimentin expression, suggesting that TNF-α does not induce EMT. TGF-β1 alone increases N-cadherin and Vimentin, and the highest increases in N-cadherin and vimentin. Interestingly, PTX significantly reduces TGF-β1 and TGF-β1 effects on N-Cadherin and Vimentin. These results showed that PTX decreases a protein expressed in mesenchymal cells considered a marker of EMT. These proteins, play a key role in reducing the mesenchymal phenotype.
Based on the results, it can be concluded that PTX decreased EMT induced by TGF-β1, TNF-α and TNF-α +TGF-β1
- Nonetheless, the primary discovery is that PTX decreases the effect of TGF-β1 alone or their combination with TNF-α on N-cadherin and Vimentin proteins after 5 days. These data are those that support our hypothesis.
Regarding cellular homeostasis
Unfortunately, our results do not allow us to answer this question; this is an important topic for future research. Nevertheless, it is we must emphasize that our cell populations had cell viability close to 95% after 5 days of treatment, so we assume that the cells maintained a stable and relatively constant internal environment, thus preserving cellular homeostasis.
- 3) As mentioned in comment 2, the authors claim that Pentoxifylline inhibits cell migration induced by TGFb1 and TNFa. Given the effects Pentoxifylline has on cell proliferation (Figure 3), the phenotype observed in Figure 4 could be due to the impairment of cell proliferation. Thus, further experiments are required to discriminate between cell proliferation and migration.
This is a good observation. There may be such a possibility; we now mention it in our document. It is important to note that the same number of cells are placed in the transwell in the invasion assay. This invasion assay measures the cellular ability to penetrate a physical barrier in 24 h; it is the active migration of the cell.
Based on the data, there are two potential explanations to consider.
The first possibility is that the antiproliferative impact of PTX could impact migration.
The second possibility is that TNF-α and TGF-β1 have not stimulated migration

Reviewer 3 Report
This paper is not very well written and describes the inhibitory effect of PTX on EMT in a cell line known as CaSki; the suppression of TNFa, TGFb, NFKB and serpine1 would appear to be involved. A major issue for me was that only one cell line was tested, usually this type of experiment would use maybe 4-5 cell lines. For those not working directly in the field it would also be useful to state the tissue of origin of the cell line in the Abstract.
Particular points
1. Lines 70/71 are a nonsense: ‘PTX has been proven to significantly increases tumor cell apoptosis 24-26 as well as in children with acute lymphoblastic leukemia’ Re-write - ……. to significantly increase tumor cell apoptosis, for example in pediatric ALL.
2. There are numerous other instances where some rewriting is required, e.g. lines 84, 291, 292, 306, 307, 343, 349, 352, 390, 473, 551.
3. Lines 434, 466, 522, 541, 548, 569, 570, 608 and probably others, the last number should be in superscript to indicate to the power of 10.
4. Fig. 1 – what does ‘undyed’ mean?
5. Lines 111 and 116 refer to Fig. 1, should be Fig.2
6. Results described on lines 116-129 do not refer to the relevant Figs.
7. Line 176 – spped
8. Explain what Serpine1 encodes (PAI-1), and why its down-regulation contributes to loss of EMT
9. Did the authors note a change in cell morphology as well as loss of markers?
10. Lines 172, 174 – the cells don’t contract, they are less migratory.
11. Unsure of reference author citation before et al, but some have 30 authors. Also check for missing page numbers, some have just volume number e.g.19, 25, 44.
Author Response
Thank you for taking the time to provide me with detailed and constructive feedback. Your comments were extremely helpful and have significantly improved our work.
Major concerns.
Usually, this type of experiment would use maybe 4-5 cell lines.
- We intended to study 3 cell lines HeLa, SiHa, and CasKi. However, we had to eliminate HeLa and SiHa cells from our study due to specific observations. We found that HeLa cells expressed a higher level of Vimentin and did not express E-cadherin. Similarly, SiHa cells barely expressed E-cadherin, and it was impossible to induce an increase in Vimentin.
- The CasKi cells express E-cadherin, making them ideal for studying the loss of the epithelial phenotype. Moreover, has been published that inflammatory stimuli such as EGF [2] IL-9 [3], and TGF-β1 [4], can induce the expression of N-cadherin and vimentin in these cells. This characteristic of CasKi cells makes them a suitable cell line to study the induction of EMT by proinflammatory cytokines and evaluate the efficacy of anti-inflammatory drugs. Several studies have focused on using CasKi cells to investigate the induction of EMT by TNF-α and TGF-β1.
- Lei, C.; Wang, Y.; Huang, Y.; Yu, H.; Huang, Y.; Wu, L.; Huang, L. Up-regulated miR155 reverses the epithelial-mesenchymal transition induced by EGF and increases chemo-sensitivity to cisplatin in human Caski cervical cancer cells. PLoS One 2012, 7, e52310, doi:10.1371/journal.pone.0052310.
- Zheng, N.; Lu, Y. Targeting the IL-9 pathway in cancer immunotherapy. Hum Vaccin Immunother 2020, 16, 2333-2340, doi:10.1080/21645515.2019.1710413.
- Li, M.Y.; Liu, J.Q.; Chen, D.P.; Li, Z.Y.; Qi, B.; Yin, W.J.; He, L. p68 prompts the epithelial-mesenchymal transition in cervical cancer cells by transcriptionally activating the TGF-β1 signaling pathway. Oncol Lett 2018, 15, 2111-2116, doi:10.3892/ol.2017.7552.
- Publications that explore the induction of EMT using only one cell line.
- Huang, Y.-H.; Chen, H.-K.; Hsu, Y.-F.; Chen, H.-C.; Chuang, C.-H.; Huang, S.-W.; Hsu, M.-J. Src-FAK Signaling Mediates Interleukin 6-Induced HCT116 Colorectal Cancer Epithelial–Mesenchymal Transition. Int. J. Mol. Sci. 2023, 24, 6650. https://doi.org/10.3390/ijms24076650
- da Costa, K.M.; Freire-de-Lima, L.; da Fonseca, L.M.; Previato, J.O.; Mendonça-Previato, L.; Valente, R.d.C. ABCB1 and ABCC1 Function during TGF-β-Induced Epithelial-Mesenchymal Transition: Relationship between Multidrug Resistance and Tumor Progression. Int. J. Mol. Sci. 2023, 24, 6046. https://doi.org/10.3390/ijms24076046
- Huang, Y.-H.; Chen, H.-K.; Hsu, Y.-F.; Chen, H.-C.; Chuang, C.-H.; Huang, S.-W.; Hsu, M.-J. Src-FAK Signaling Mediates Interleukin 6-Induced HCT116 Colorectal Cancer Epithelial–Mesenchymal Transition. International Journal of Molecular Sciences 2023, 24, 6650.
For those not working directly in the field it would also be useful to state the tissue of origin of the cell line in the Abstract.
- This is corrected.
Particular points
- Lines 70/71 are a nonsense: ‘PTX has been proven to significantly increases tumor cell apoptosis 24-26 as well as in children with acute lymphoblastic leukemia’ Re-write - ……. to significantly increase tumor cell apoptosis, for example in pediatric ALL
This is corrected.
- There are numerous other instances where some rewriting is required, e.g. lines 84, 291, 292, 306, 307, 343, 349, 352, 390, 473, 551.
This is corrected.
- Lines 434, 466, 522, 541, 548, 569, 570, 608 and probably others, the last number should be in superscript to indicate to the power of 10.
This is corrected.
- Fig. 1 – what does ‘undyed’ mean?
There was a mistake in the graph where "undyed control" referred to cell autofluorescence. To prevent any confusion, it has been removed.
- Results described on lines 116-129 do not refer to the relevant Figs.
This is corrected.
- Lines 111 and 116 refer to Fig. 1, should be Fig.2
This is corrected.
- Results described on lines 116-129 do not refer to the relevant Figs.
This is corrected.
- Line 176 – spped
This is corrected.
- Explain what Serpine1 encodes (PAI-1), and why its downregulation contributes to loss of EMT.
This information has been integrated into the document.
- Did the authors note a change in cell morphology as well as loss of markers?
- We noticed a slight change in the morphology of CasKi cells, but it wasn't particularly impressive. Some changes were observed in the groups treated with TGF-α and TNF-α +TGF-β1. We found that PTX can reverse these changes. We have attached images of the groups taken 5 days after treatment.
- Lines 172, 174 – the cells don’t contract, they are less migratory.
This is corrected.
- Unsure of reference author citation before et al, but some have 30 authors. Also check for missing page numbers, some have just volume number e.g.19, 25, 44.
This is corrected.

Round 2
Reviewer 1 Report
All my comments and questions from the previous round were properly addressed.
Author Response
Thank you for taking the time to provide me with detailed and constructive feedback. Your comments were extremely helpful and have significantly improved our work.
Reviewer 2 Report
The authors did not convincingly addressed my comments on migration. To my opinion, scratch assays are not able to discriminate between proliferation and migration. Recently, a new method has been described to really address drug effects on migation while checking the proliferation (PMID: 37055469). I invite the authors to have a look at it and maybe implement it to really demonstrate the effect on migration.
Despite a progression in writing, the paper is still poorly written and has to be improved.
Author Response
Response to Reviewer 2 Comments
Point 1: The authors did not convincingly address my comments on migration. In my opinion, scratch assays cannot discriminate between proliferation and migration. Recently, a new method has been described to address drug effects on migration while checking proliferation (PMID: 37055469). I invite the authors to look at it and maybe implement it to demonstrate the effect on migration.
Thank you for your recommendation; it is interesting how you determined the difference between cell proliferation and migration; however, at this time, we do not have a collaborator who has the necessary knowledge in Python to replicate the analysis shown in the recommended article. It gave us several ideas to compare proliferation and surface change of the wounds in the images (migration) using ImageJ software; to determine proliferation VS migration in our work at your discretion and evaluation.
Methodology:
Determination of the proliferation index (PI)
Context: We determined proliferation by SRB staining technique at 3 days (results not shown in the article) and at 5 days (Figure 3), where we analyzed the absorbance of SRB of each treatment group by spectrometry.
- We related the absorbance of the groups at 3 days to that of the groups at 5 days of proliferation, where the absorbance of the groups at 3 days represents 1 (100%), and equation (1) was used to determine the increase in proliferation of each group in the index during the 48h between the two experiments. This relation was taken and adapted from [1], while the relationship between the absorbance of a dye such as SRB to determine proliferation is shown in [2].
PI= ( Ab5/Ab3 -1)
Equation 1 (modified from ref (1).
Where:
PI = Proliferation index
Ab5 = Absorbance at 5 days of proliferation
Ab3 = Absorbance at 3 days of proliferation
Determination of the migration index
Context: For the wound closure assay, the cells were treated with the stimuli mentioned above for 5 days, then the medium was removed and washed 3 times with PBS to remove the treatments, the wound was made, and fresh medium was added without treatments to continue its culture, then photographs were taken at 0, 6, 12 and 24h after wounding to quantify the surface area of the wounds at different times with ImageJ software.
1) For the determination of MI, we first elucidated the area that the cells traveled from the initial wound to the wound at time t (6, 12, or 24h) with equation (2).
AC(t) = Ai - At
Equation 2 ( Modified from ref (3)
Where:
AC(t) = Area closed at time t
Ai = Initial wound area
At = Final wound area at time t
2) Then, using the ImageJ software, the average surface area occupied by each cell in the wound closure images at 24h was calculated similarly to the reported method [4], only that this time the manual way was used in which the following steps were followed: open - choose pencil - draw the outline of a cell - measure - record the area. The area of 10 cells per image was taken according to each experimental group, and the results were averaged.
3) Equation 3 calculated the number of cells that migrated, closing the wound in a determined time (6, 12, and 24h).
Nc(t) = AC(t) / CSA
Equation 3
Where:
Nc(t) = Number of migratory cells at time t
AC(t) = Area closed at time t
CSA = cell surface area
4) Once the number of migrating cells that closed the wound (Nc) was calculated, the migration index (MI) was determined. For this, the results of the Nc of each group at 12h were taken as 100% (1) and compared. With the Nc results at 24h later, use equation 4 to determine the MI (how many cells migrated more at 24h compared to 12h)
MI = ( NC24h / NC12h -1)
Equation 4 (Modified from ref (1)
Where:
MI = Migration index
NC24h = Number of migratory cells at 24h
NC12h = Number of migratory cells at 12h
Integration of results
To compare the proliferation index (at 48h) and the migration index (at 12h), the value of the proliferation index was divided by 4 to obtain a theoretical value of proliferation at 12h and thus compare it with the migration index at 12h (Figure 1 1) and 24h (Figure 2), the results are as follows.
Figure 1. Comparison of proliferation index (PI) VS migration index (MI) at 12h. Data plotted were from three independent trials in triplicate. Statistical analysis was performed with the U-Mann Whitney test with a statistical significance of P < 0.05.
Figure 2. Comparison of proliferation index (PI) VS migration index (MI) at 24h. Data plotted were from three independent trials in triplicate. Statistical analysis was performed with the U-Mann Whitney test with a statistical significance of P < 0.05.
Discussion
Indeed, proliferation and migration phenomena are inherent to the wound closure migration assay; however, depending on the cell line as well as the treatments, the results may be due more to one of these effects than the other; in this proposed analysis, it is shown that PI is lower than MI in the different experimental groups in CaSki cells, therefore, with this proposed analysis, we suggest that the results in Figure 4 may be mainly due to cell migration than cell proliferation.
References
- Munson, M.E. An improved technique for calculating relative response in cellular proliferation experiments. Cytometry Part A 2010, 77A, 909-910, doi:https://doi.org/10.1002/cyto.a.20935.
- Venter, C.; Niesler, C. Rapid quantification of cellular proliferation and migration using ImageJ. BioTechniques 2019, 66, 99-102, doi:10.2144/btn-2018-0132.
- Ammann, K.R.; DeCook, K.J.; Li, M.; Slepian, M.J. Migration versus proliferation as contributor to in vitro wound healing of vascular endothelial and smooth muscle cells. Exp Cell Res 2019, 376, 58-66, doi:10.1016/j.yexcr.2019.01.011.
- Baviskar, S.N. A Quick & Automated Method for Measuring Cell Area Using ImageJ. The American Biology Teacher 2011, 73, 554-556, doi:10.1525/abt.2011.73.9.9.

Reviewer 3 Report
The authors still only use the one cell line. It is the one that fits their hypothesis, the other two do not so they ignore them!
Author Response
Response to Reviewer 3 Comments
Point 1: The authors still only use the one cell line. It is the one that fits their hypothesis, the other two do not so they ignore them!
Point 1:
- HeLa and SiHa cells were not ignored; they were not good studio models. It was not possible to induce EMT. At the protein level, the proportion of EMT state was variable; for example, SiHa cells in basal conditions show low E-cadherin, N-cadherin, and Vimentin expression.
- In HeLa cells, rapidly proliferated, formed spheroids and show the presence of mesenchymal markers, like snail, vimentin, and N-cadherin, and the decrease in epithelial marker E-cadherin suggests that HeLa cells are mesenchymal. Moreover, HeLa cells present a fibroblast-like spindle appearance, further confirming their mesenchymal nature. HeLa was highly mesenchymal. Also, the TGF-β long exposition in HeLa cells has caused cell death. The HeLa cell is not a good model for studying EMT. Y Pang et al 2018 Biofabrication 10 044102. DOI 10.1088/1758-5090/aadbde.
- The objective of the EMT induction in these cell lines was not possible because HeLa and SiHa cell lines possess various cellular features associated with a mesenchymal cell state.
- CasKi cells in basal conditions don't show mesenchymal proteins. In addition, CasKi cells serve as a model for the study of advanced cervical carcinoma. J Gen Virol. 1994 May;75 (Pt 5):1139-47. doi: 10.1099/0022-1317-75-5-1139.
- Our focus of this study was to explore the effects of PTX on epithelial-mesenchymal transition change in a model of EMT in CC.
- There is a large number of publications exploring EMT using a single-cell line. Only in 2023, 110 publications used CasKi cells.Publications that explore the induction of EMT using only one cell line. Finally, we think that our study using the CasKi cell line represents a good in vitro model to evaluate EMT and its changes in response to different therapeutic schemes.
-
- Example:
- Huang, Y.-H.; Chen, H.-K.; Hsu, Y.-F.; Chen, H.-C.; Chuang, C.-H.; Huang, S.-W.; Hsu, M.-J. Src-FAK Signaling Mediates Interleukin 6-Induced HCT116 Colorectal CancerEpithelial–Mesenchymal Transition. J. Mol. Sci. 2023, 24, 6650. https://doi.org/10.3390/ijms24076650
- da Costa, K.M.; Freire-de-Lima, L.; da Fonseca, L.M.; Previato, J.O.; Mendonça-Previato, L.; Valente, R.d.C. ABCB1 and ABCC1 Function during TGF-β-Induced Epithelial-Mesenchymal Transition: Relationship between Multidrug Resistance and Tumor Progression. J. Mol. Sci. 2023, 24, 6046. https://doi.org/10.3390/ijms24076046
